# Compensatory Structural Growth Responses of Early-Succession Native Warm-Season Grass Stands to Defoliation Management

**Vitalis W. Temu** * and **Maru K. Kering**

Agricultural Research Station, Virginia State University, Petersburg, VA 23806, USA; mkering@vsu.edu
* Correspondence: vtemu@vsu.edu; Tel.: +1-804-524-6717; Fax: +1-804-524-5186

**Abstract:** There is a growing recognition of the significance of unique morphological and physiological adaptation of native warm-season grasses (NWSG) of North America as summer forage resources and major grassland ecosystem components. Defoliation management plays a major role in ensuring eco-friendly utilization of grassland natural resources. To assess swonard structural responses of big bluestem (BB, *Andropogon gerardii* Vitman), eastern gamagrass (GG, *Tripsacum dactyloides* L.), indiangrass (IG, *Sorghastrum nutans* L. Nash), and switchgrass (SG, *Panicum virgatum* L.) stands to seasonal changes in harvest regimes, a five-year forage harvesting trial was conducted, in a randomized complete block design, at Virginia State University's research farm. Vegetation structural response attributes (sward-height, canopy closure, stand density and basal cover) of newly established the NWSG stands to second year changes in harvest regimes were monitored. In 2013, 64 plots of year-old stands of transplanted BB, GG, IG, and SG separated by ≥120-cm alleys were cut once in early-August and mid-November to suppress weeds and promote tillering. Starting June 2014, each plot had three 1.5-m wide side-by-side harvest-strips cut once-, twice-, or thrice year$^{-1}$ (frequencies) ending mid-Oct for four consecutive years followed by a single mid-summer harvest in 26 June 2018, using a forage plot-harvester. In 2015, harvest frequencies for the three- and single-cut strips, in plots 32–64, were switched/flipped once and never reverted. Data was recorded on four early-summer and late-fall sward heights, from each strip at 60-cm intervals before the first and the last harvest, each year. early-spring basal- and canopy-diameter, for mid-April 2015 and 2016, concurrent early-spring canopy light interception, using the LI-191 Line Quantum Sensor, and season-end visual obstruction heights, for stand density in 2016 and 2017. All regrowth sward-heights showed effects of harvest frequency and exhibited compensatory structural responses to the change in harvest regimes. Basal and canopy diameters tended to be greater for the single-cut strips that were previously cut thrice.

**Keywords:** defoliation frequency; forage; sward-height; structure; canopy; light interception; habitat; warm-season; big bluestem; gamagrass; indiangrass; switchgrass; compensatory

## 1. Introduction

A growing recognition of the unique suitability of native warm-season grasses (NWSG) of North America for various economic and ecological uses has generated interests in management strategies for sustainable utilization. As forage plants, most NWSGs often grow better and persist longer under hot and drought growing conditions than their exotic counterparts such as bermudagrass (*Cynodon dactylon* L. Pers), bahiagrass (*Paspalum notatum* Flueggé), and dallisgrass (*Paspalum dilatatum* Poir.). Being morphologically and physiologically tolerant to harsh growing conditions also makes them suitable candidates for various ecosystem services such as wildlife habitat, soil conservation, bioenergy production, carbon sequestration, stabilizing stream banks, riparian buffers and filtering off sediments from runoff waters in agricultural landscapes. In many ways, NWSGs have shown the potential to play a role towards alleviating some major global challenges of

food and/energy insecurity, climate change and environmental sustainability. Thus far, five mostly researched NWSG species—big bluestem (BB, *Andropogon gerardii* Vitman), eastern gamagrass (EG, *Tripsacum dactyloides* L.), indiangrass (IG, *Sorghastrum nutans* L. Nash), little bluestem (LB, *Schizachyrium scoparium* Michx. Nash), switchgrass (SG, *Panicum virgatum* L.) are promising and have shown desirable attributes for this multiple use [1]. Regardless of the long established economic and environmental benefits of NWSGs in managed grasslands, their adoption remains very low. Hindrances to their incorporation into forage systems, for example, are mostly associated with high costs of establishment and inability to withstand grazing when compared with conventional/exotic forage grasses [2]. The inability for farmers to realize the well-promoted unique ecological advantages that NWSGs have over the conventional forage grasses remains a challenge.

In many aspects, these deterrents to faster incorporation of NWSGs into forage systems seem mostly rooted in misperceptions around their stand persistency, forage quality and proneness to weeds pressure and bush encroachment. These are, usually, negative impacts of severe defoliation and deferential growth responses of the plant components and also reduction in leaf area coupled with changes in respiratory and growth rates, as well as carbon allocation patterns [3]. Because of their unique morphological features and physiological characteristics, NWSG stands require appropriate defoliation management to sustain high forage productivity. To be appropriate, decisions on the timing, intensity, and frequency of defoliation events must be mindful of their potential effects to critical plant growth responses. For tall-growing bunch grasses, defoliation usually removes the hormonal dominancy of reproductive tillers and that stimulates bud break and production of the relatively leafier vegetative tillers [4,5]. The rate at which plants may repair the damaged tissues and restore their lost photosynthetic capacities is dependent on the proportion of functional leaf area retained and timing and the time between successive defoliation events [3,6–8].

Depending on the management objective(s), mainly, biomass production, soil conservation and/or ecosystem services, relevant responses may include changes in tiller numbers, stand density, vegetation structure, species composition and spatial distribution, and growth performance amongst others. Proper defoliation management of NWSG stands must allow enough time for recovery growth and also minimize excessive losses of or damages to growing points, which happen to be high above the ground and mostly beyond the common cutting heights [3,9,10]. As an adaptation to grazing, regrowth of defoliated NWSGs usually involves apportioning more resources towards faster regrowth and repair or even compensate for sustained physical damages or loss of meristematic tissues [11]. However, species morphological adaptations to tissue damage differ and are reflected in a plants' tolerance to defoliation and respective growth responses [12–14]. The degree to which plants may compensate for tissue damages or loss is also influenced by local factors such as; precipitation, defoliation intensity and history [15–18]. Therefore, appropriate defoliation management of mixed stands must also consider how the most dominant species may respond to stresses induced under the prevailing weather conditions.

Where grassland management is interested more in ecosystem services, monitoring the stand responses to defoliation need to focus on changes in vegetation structure, species composition and the proportion of ground covered by plant biomass. Vegetation structural changes include the Canopy architecture, which affects the physiological functions of forage plants, the forage quality on offer to grazing animals [19], such processes as photosynthesis, transpiration, cell enlargement, and species competition in a plant community, root growth, soil moisture retention, soil health and residue decomposition [20]. For a plant community, canopy structure is the spatial arrangement of the above-ground organs in it, which includes leaves and other photosynthetic organs as well as the stems and branches for their support and strategic positioning. Commonly used parameters for monitoring changes in vegetation structure include canopy and basal diameters [21], sward-height that is mostly preferred in the management of grazing system due to its strong influence on bite dimensions [22] and canopy light interception, which varies positively with leafiness and

forage quality [23]. Canopy structure has indirect influence on soil moisture and temperature, affects root growth, soil moisture retention, biomass residue composition, and other soil microbial processes [20,24]. These canopy structural attributes have implications on species composition, ground-level air temperatures and suitability of the stand for wildlife habitat. In a grazing system, canopy architecture affects both the physiological functions of forage plants and their forage quality [19].

Another parameter is visual obstruction, which is a non-destructive tool for estimating herbaceous standing crop in tallgrass prairie [25]. As a technique for monitoring standing crop and vegetation height and density on grasslands, visual obstruction, which integrates plant-height and density, is simple, cost effective, and provides pertinent information for both livestock and wildlife management purposes [26]. In plant communities, these structural changes have impact on species composition whose functional traits influence the efficiency of resource use by the biological system [27]. The resulting spatial and temporal changes in sward structure also have implications on their wildlife habitat qualities for different types of grassland birds and small mammals. For ground-nesting birds, the most habitat quality features include visibility while foraging, vigilance to predators, concealment, and easy mobility through the stands [28,29]. While a second-year switching of harvest frequencies between the single- and three-cut NWSG stands have exhibited compensatory forage yield responses [30], information on its associated impact on wildlife habitat quality is not clear. This study, therefore, was established to generate data that may help in developing sustainable defoliation management strategies for young NWSG stands for dual use summer forage production and wildlife habitat. The study focused on NWSGs' sward height, canopy closure, stand density, and ground cover attribute responses to a second year change in harvest frequencies. It was hypothesized that reducing the number of harvests per year would result in faster regrowth rates enough to compensate for prior-year losses in stand vigor that will also reflect in vegetation structure.

## 2. Materials and Methods

### 2.1. Location and Field Preparations

The study was conducted at Randolph Farm–Virginia State University's research and demonstration farm located in Chesterfield county, Virginia at 37°13′43″ N; 77°26′22″ W, about 45 m above sea level. The soil at the farm is Bourne series fine sandy loam (mixed, semiactive, thermic Typic Fragiudults) with low organic matter content. No fertilizers were applied to the NWSGs for the duration of this study. By the summer of 2013, the study area had a 20-year June, July, and August average precipitation of 92, 113, and 121 mm with day temperatures of 30.2, 32.1, and 31.2 °C, respectively [31]. The mean monthly temperatures and precipitation amounts around the study area from April through October for the 2012 through 2018 production years are shown in Figures 1a and 1b, respectively. In 2013, 64 plots (roughly 6-m W × 7-m L) of year-old transplanted BB, GG, IG, and SG stands separated by ≥120-cm alleys received a first cut in early-August and second one in mid-November to suppress annual weeds. The seedlings were raised in the greenhouse from seeds, U.S. ecotype for GG and NC ecotypes for the others, supplied by Ernst Conservation Seeds Inc., Meadville, PA, USA. At planting, the seedlings were spaced 30 × 45 cm within and between rows, respectively. The plots were in eight 8-plot rows, in which each NWSG species was assigned two, in a randomized complete block design. Starting early-June of 2014, three 1.5-m wide side-by-side strips in each plot were cut once-, twice-, or thrice per year (harvest frequencies). After four consecutive forage harvesting years, a fifth-year single mid-summer harvest followed on 26 June 2018. Harvest dates for the entire study are summarized in Table 1, by year and treatment. A CIBUS F Plot Forage Harvester (Wintersteiger Ag, Dimmelstrasse, Austria) with a 120-cm cutting width cutting height set at 18-cm was used. During the first through fourth year, harvesting for the three-cut systems happened in early-June, late-July to early-Aug, and late-September to mid-October. In most cases, the same first and last harvest dates were used for the two-cut systems except when weather and logistical problems impacted operations. The last harvest date for the

three- and two-cut systems was also basically the same for the one-cut per year system. In the second harvest-yearn, however, the harvest frequencies were flipped once for the single- and three-cut strips in plots 33–64, but not in the other 32 plots (Figure 2). The flipping of harvest frequencies was never reverted throughout the study. Having the flipped and not flipped plots in separate blocks was necessary to avoid potential shading effect of single-cut strips in one row on the three-cut strips of a neighboring row. For the same reason, the middle alley separating the two blocks was also made 60-cm wider than the 120-cm alleys separating plots, within blocks. To facilitate machine operations on different scheduled harvest dates and to limit associated disturbance to the harvest strips, end-to-end aligned strips within each 8-plot row were assigned the same harvest frequency.

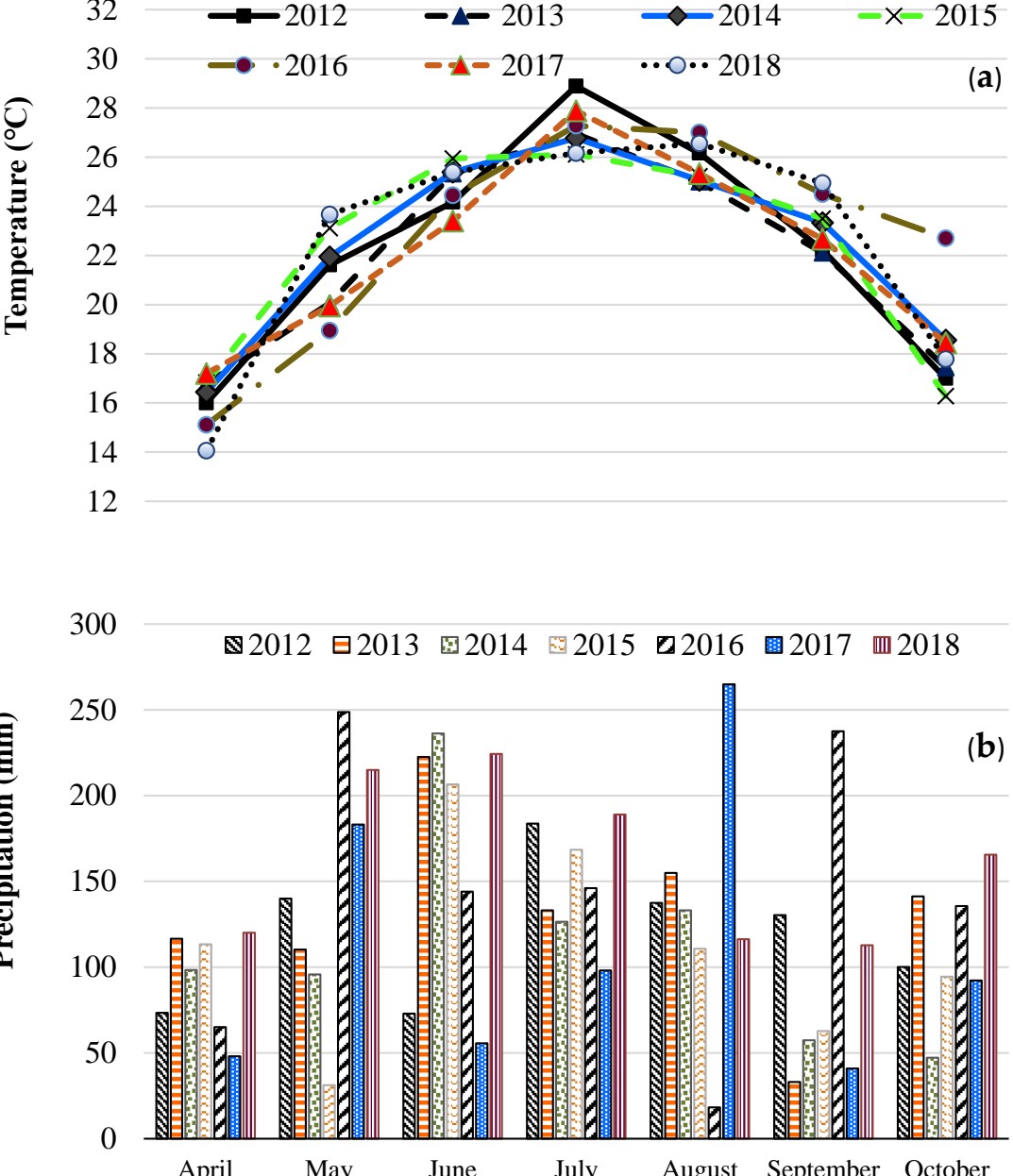

**Figure 1.** Mean monthly temperatures (**a**), top, and precipitation (**b**), below, recorded at nearby National Weather Service Stations (Hopewell and Petersburg, VA, USA) from April through October during the 2012–2018 production years. Adopted from Temu, et. al. (2022) [29].

**Table 1.** Actual harvest dates for three different cutting frequencies (cuts year$^{-1}$) on newly established NWSG stands recorded from 2013 to 2018.

| | | Harvest Dates by Harvest Regime | | |
|---|---|---|---|---|
| Year | Cuts | Three Cuts | Two Cuts | Single Cut |
| 2013 * | 1st | 5 June 2013 | 5 June 2013 | 5 June 2013 |
| | 2nd | 18 November 2013 | 18 November 2013 | 18 November 2013 |
| 1st (2014) | 1st | 24 June 2014 | 23 July 2014 | 14 September 2014 |
| | 2nd | 29 July 2014 | 12 September 2014 | |
| | 3rd | 12 September 2014 | | |
| 2nd (2015) | 1st | 14 June 2015 | 18 June 2015 | 14 October 2015 |
| | 2nd | 31 July 2015 | 14 October 2015 | |
| | 3rd | 14 October 2015 | | |
| 3rd (2016) | 1st | 18 June 2016 | 18 June 2016 | 24 October 2016 |
| | 2nd | 05 August 2016 | 24October 2016 | |
| | 3rd | 24 October 2016 | | |
| 4th (2017) | 1st | 19 June 2017 | 29 June 2017 | 18 October 2017 |
| | 2nd | 17 August 2017 | 18 October 2017 | |
| | 3rd | 17 October 2017 | | |
| 5th (2018) * | N/A | 26 June 2018 | 26 June 2018 | 26 June 2018 |

* All plots experienced the same pre- and post-treatment harvests in 2013 and 2018, respectively; N/A = Not applicable.

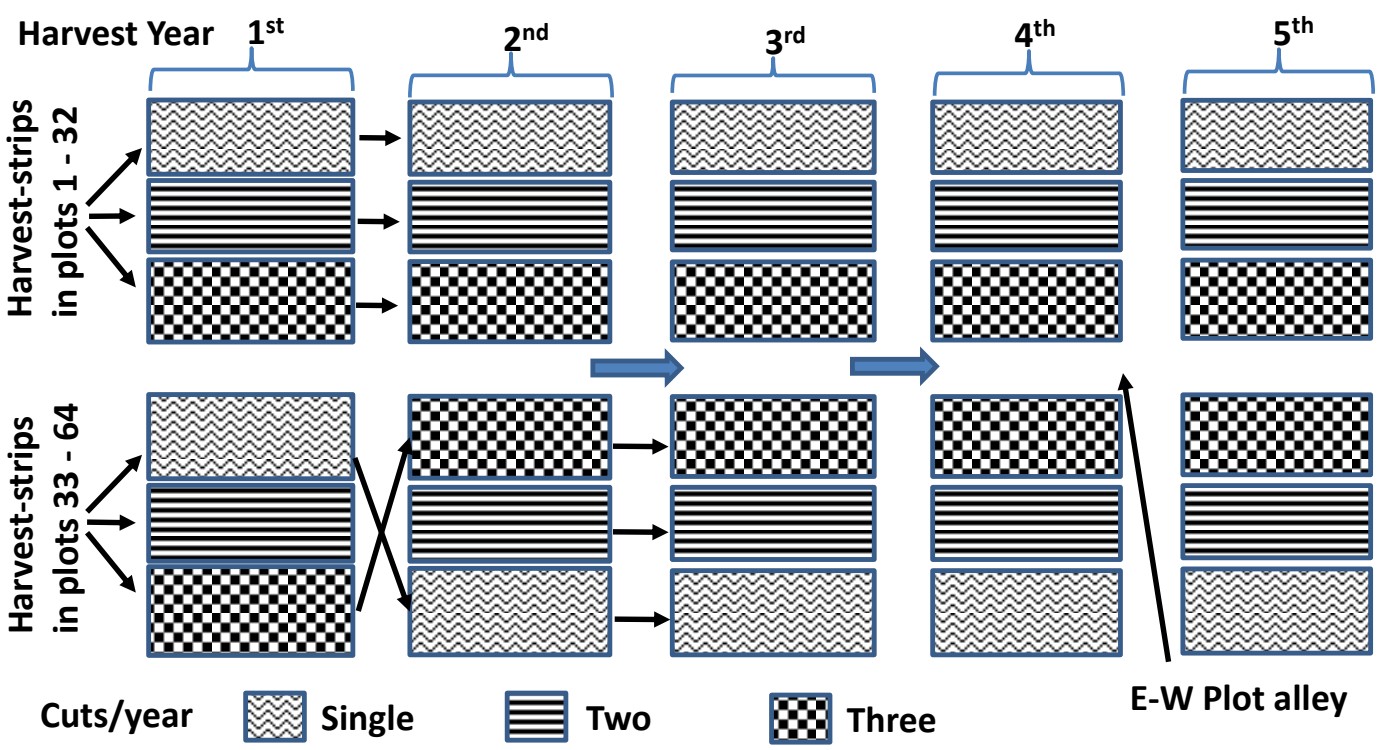

**Figure 2.** Relative arrangement of harvest-strips within a plot assigned to one, two, or three cuts/year, with harvest regimes being flipped between the single- and three-cut strips after the 1st year–2014, for plots 32–64 and all plots receiving a single late-June harvest in the 5th year–2018, adopted from Temu, et. al. (2022) [29].

*2.2. Vegetation Measurements*

A day or two before the first and the last harvest of the year, four early-summer and late-fall sward height (cm) measurements were recorded from every harvest strip at 60-cm

intervals. A sward height reading was recorded as the highest point above ground at which a meter stick, held horizontally above the sward and perpendicular to a vertical Robel pole, touched at least two native grass leaves on separate rows. For the data analysis, the sward-height readings were entered as four-point averages. Around mid-April of the 2015 and 2016 harvest years, NWSG early-spring basal diameter (BD) measurements at about 2.5 cm above soil surface (crown's widest cross-sectional distance, cm) of three inner-row stubbles $\geq$ 1-m apart were recorded from each harvest strip. About a month later, three respective canopy diameter (CD) measurements (widest horizontal distance between canopy edges) of inner-row clumps perpendicular to the plot length orientation, spaced $\geq$ 1-m apart, were also recorded. From the measured BD and CD values, a CD:BD ratio was calculated. Due to gradual merging of crowns, within rows, individual BD and CD readings for the fourth and fifth harvest years were considered misleading in assessing response to the second-year flipping of harvest frequencies.

The treatment effects assessment was also done on canopy closures in the regrowth stands, during both 2015 and 2016. For that, early-spring instantaneous photosynthetically active solar radiation (PAR) intercepted by the vegetation layer in each harvest-strip was recorded between the 12:00 and 14:00 h. The day-time for the PAR readings was intended to minimize likely distortions that the recorder's shadow or tall plants in neighboring strips might have on the actual proportions of the intercepted solar radiation. For each PAR reading above the canopy (PARa), five matching readings, $\geq$1-m apart, of that reaching the ground surface beneath (PARb) were also recorded. From the PAR readings, average light interception was calculated as = $\sum$ [(PARa $-$ PARb)/PARa] $\times$ 100/5, [32]. The PAR measurements, $\mu$mol m$^{-2}$ s$^{-1}$, were taken using the LI-191 Line Quantum Sensor (LI-COR 2000, LICOR, Lincoln, NE, USA).

During the 2016 and 2017 growing seasons, Visual Obstruction height (VOH) measurements as indicators of how the second harvest-year change in harvest frequencies may have influenced the subsequent stand density, that also has implications on visibility-related wildlife habitat qualities, was also done. As a technique, VOH is simple, cost effective, and provides pertinent information to both livestock and wildlife management purposes [26]. The VOH method is also considered an effective, non-destructive tool for estimating herbaceous standing crop in tallgrass prairies [25]. In the current study, VOH was recorded as the height at which a naked eye on the opposite end of the 6-m long harvest-strip could sight the lowest unobstructed mark on a graduated pole through the stand, when viewed at about a meter above the ground. Concurrently, canopy height measurements were also recorded using a modified Robel pole and a meter rule.

*2.3. Data Analysis*

The data were organized and subjected to analysis of variance (ANOVA) as a RCBD with, the year of assessment, defoliation management (system), species, and harvest frequency (cuts) as fixed effects. For the analysis, the windows-based SAS software 9.4 (SAS Institute Inc., Cary, NC, USA) was used. Because of significant year, species and treatment interactions, multiple ANOVA procedures were done for each year separately to compare treatments within species and harvest regimes. The ANOVA first compared the single-, two-, and three-cut systems on their sward structure attributes within year, species and harvest regimes (flipped or same). Then, a separate ANOVA compared the harvest regimes within year, species and harvest frequency. Respectively, the probabilities of difference from each run were used for the means comparison within year, species and harvest frequency or harvest regime. Means were compared by the Fisher's Least Significant Difference test at $\alpha$ = 0.05.

**3. Results and Discussion**

The ANOVA results showed highly significant ($p < 0.001$) main effects of year, species and harvest frequency as well as their two- and three-way interactions on the measured sward heights (Table 2).

**Table 2.** The ANOVA F and *p* values for the main and interaction effects of Year, harvest regime (System), Species, and harvest frequency (Cuts) on early- and late-season sward heights of young native warm-season grass stands harvested once, twice, and thrice year$^{-1}$ recorded from 2014 to 2018.

| | Early-Season Heights | | | Late-Season Heights | | |
|---|---|---|---|---|---|---|
| Source | DF | Fα | *p* > Fα | DF | Fα | *p* > Fα |
| Model | 119 | 49.63 | <0.001 | 95 | 300.75 | <0.001 |
| Year | 4 | 133.37 | <0.001 | 3 | 452.26 | <0.001 |
| System | 1 | 9.60 | 0.002 | 1 | 0.55 | 0.458 |
| Year × System | 4 | 13.37 | <0.001 | 3 | 35.95 | <0.001 |
| Species | 3 | 727.32 | <0.001 | 3 | 1562.85 | <0.001 |
| Year × Species | 12 | 98.54 | <0.001 | 9 | 24.93 | <0.001 |
| System × Species | 3 | 6.93 | 0.001 | 3 | 17.14 | <0.001 |
| Year × System × Species | 12 | 8.41 | <0.001 | 9 | 3.83 | <0.001 |
| Cuts | 2 | 551.60 | <0.001 | 2 | 8889.27 | <0.001 |
| Year × Cuts | 8 | 38.16 | <0.001 | 6 | 215.31 | <0.001 |
| System × Cuts | 2 | 13.41 | <0.001 | 2 | 26.11 | <0.001 |
| Year × System × Cuts | 8 | 2.21 | 0.025 | 6 | 1.38 | 0.218 |
| Species × Cuts | 6 | 32.83 | <0.001 | 6 | 338.78 | <0.001 |
| Year × Species × Cuts | 24 | 6.56 | <0.001 | 18 | 46.20 | <0.001 |
| System × Species × Cuts | 6 | 0.71 | 0.641 | 6 | 5.83 | <0.001 |
| Year × System × Species × Cuts | 24 | 0.49 | 0.982 | 18 | 4.27 | <0.001 |
| Error | 840 | | | 672 | | |
| Corrected Total | 959 | | | 767 | | |

DF = degrees of freedom; *p* > Fα = probability of difference between means within species.

As well, ANOVA for the two- and three-cut systems showed highly significant main effects of year, species, harvest frequency, and harvest timing on light interception, visual obstruction, as well as basal and canopy diameters (Table 3). The interaction effects, however, were equally strong for some attributes but not observed on others.

**Table 3.** The ANOVA *p* values for the main and interaction effects of Year, harvest regime (System), Species, and harvest frequency (Cuts) on the percent light interception (PARi), basal diameter (CD), canopy diameter (BD) readings and CD:BD ratio recorded in May 2015 and 2016, and visual obstruction heights (VOH) recorded in October 2016 and 2017, from young native warm-season grass stands harvested once, twice, and thrice year$^{-1}$.

| Source | DF | *p* > Fα | | | | |
|---|---|---|---|---|---|---|
| | | PARi | VOH | BD | CD | CBDR |
| Model | 47 | <0.001 | <0.001 | <0.001 | <0.001 | <0.001 |
| Year | 1 | <0.001 | 0.003 | <0.001 | <0.001 | <0.001 |
| System | 3 | <0.001 | <0.001 | <0.001 | 0.001 | 0.001 |
| Year × System | 3 | <0.001 | 0.153 | 0.049 | <0.001 | <0.001 |
| Species | 2 | <0.001 | <0.001 | <0.001 | <0.001 | <0.001 |
| Year × Species | 2 | 0.160 | <0.001 | <0.001 | <0.001 | <0.001 |
| System × Species | 6 | <0.001 | <0.001 | 0.937 | 0.575 | 0.151 |
| Year × System × Species | 6 | 0.229 | 0.081 | <0.059 | 0.001 | 0.002 |
| Cuts | 1 | <0.001 | 0.607 | 0.096 | <0.001 | <0.001 |
| Year × Cuts | 1 | <0.001 | 0.107 | 0.154 | <0.001 | 0.004 |
| Manage × Cuts | 3 | <0.001 | 0.058 | 0.370 | 0.575 | 0.454 |
| Year × System × Cuts | 3 | 0.020 | 0.894 | 0.699 | 0.357 | 0.832 |
| Species × Cuts | 2 | 0.014 | 0.277 | 0.041 | <0.001 | 0.041 |
| Year × Species × Cuts | 2 | 0.204 | 0.031 | <0.001 | 0.022 | 0.374 |
| System × Species × Cuts | 6 | 0.643 | 0.923 | 0.109 | 0.229 | 0.038 |
| Year × System × Species × Cuts | 6 | 0.927 | 0.249 | 0.840 | 0.715 | 0.637 |
| Error | 336 | | | | | |
| Corrected Total | 383 | | | | | |

DF = degrees of freedom; *p* > Fα = probability of difference between means within species.

Because of significant factor interactions, treatment means within-species and for each harvest regime are presented, separately, for each year.

### 3.1. Early-Summer Sward-Heights

During the 2014 growing season, the newly established NWSG stands, which had already received two common cuts in July and November 2013, experienced a second harvest year. As expected, the early-summer sward heights, which ranged from about 42 cm (BB) to 65 cm (IG), Table 4, reflected differences in the species growth responses to the common defoliation management during stand establishment. The observed sward-height differences were also consistent with two reported mechanisms used by plants to cope with herbivory; the ability to reduce their probability of being grazed or the increase in ability to recover following grazing [33]. It is also reported that a plant's response to defoliation management is related mainly to the species morphological and physiological characteristics [34]. However, the 2015 early-season sward-height records, that followed the first season under the single- two- and three-cut year$^{-1}$ harvest regime showed significant defoliation treatment effects ($p < 0.001$) (Figure 3). Within species, all strips harvested thrice the previous year (2014) had the shortest sward-heights, in 2015 that ranged from about 36.4 cm in SG to an average of 55.9 cm in GG plots.

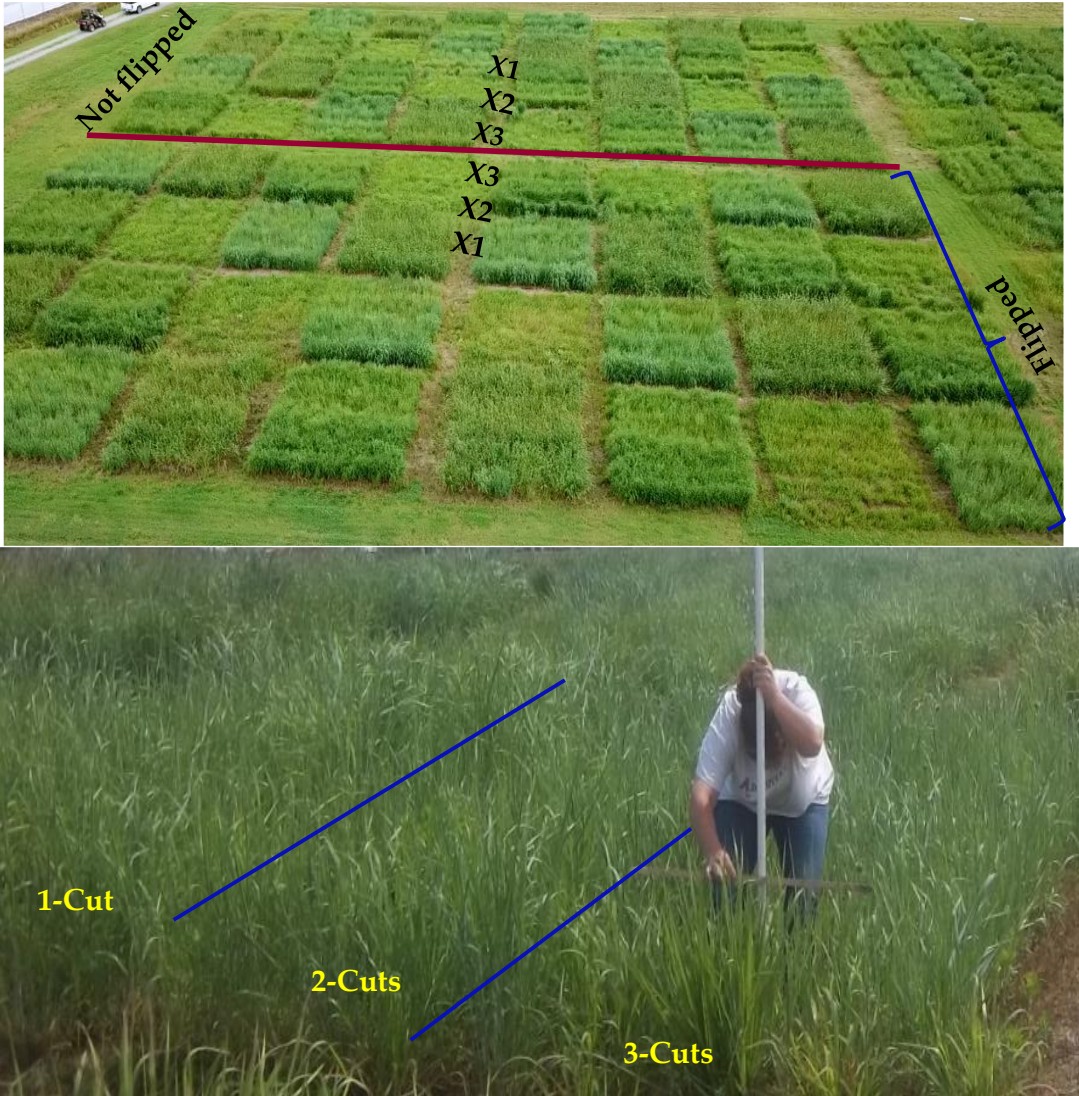

**Figure 3.** Top; Aerial view of 64 native warm-season grass plots showing relative arrangement of single-, two-, and three-cut strips (X1, X2, X3) in the block harvested the same consecutively and the one where the single- and three-cut strips were flipped during the second year. Bottom; early-summer regrowth sward-height measurement in the side-by-side harvest strips.

**Table 4.** Effects of a seasonal change (Same vs. Flipped) in harvest frequency (Cuts) on subsequent early-summer and late-fall † sward heights of young NWSG stands cut once, twice or thrice year$^{-1}$ recorded from September 2014 to October 2018.

| Year | Cuts | Species and Harvest Regime | | | | | | | |
|---|---|---|---|---|---|---|---|---|---|
| | | **Big Bluestem** | | **Gamagrass** | | **Indiangrass** | | **Switchgrass** | |
| | | Same | Flipped | Same | Flipped | Same | Flipped | Same | Flipped |
| | | Early-summer Sward Heights | | | | | | | |
| | | ----------------------------------------------cm---------------------------------------------- | | | | | | | |
| 2014 | Twice | 43.3 A§ | 42.4 A§ | 53.1 A | 56.2 A | 65.4 A | 63.8 A | 53.7 A | 50.8 A |
| | Thrice | - | - | - | - | - | - | - | - |
| 2015 | Once | 62.4 aA | 62.4 aA | 77.9 aA | 75.1 aA | 44.8 aA | 45.4 aA | 66.1 aA | 67.2 aA |
| | Twice | 52.3 bA | 52.2 bA | 69.8 bA | 63.0 bB | 44.6 aA | 43.1 aA | 42.4 bB | 45.3 bA |
| | Thrice | 45.7 cA | 45.1 cA | 55.9 cA | 57.3 cA | 39.1 bA | 38.3 bA | 36.4 cB | 39.9 cA |
| | *p* > α # | <0.001 | <0.001 | <0.001 | <0.001 | <0.001 | <0.001 | <0.001 | <0.001 |
| 2016 | Once | 67.6 aA | 65.1 aA | 82.3 aA | 80.8 aA | 47.9 aA | 49.4 aA | 73.2 aA | 67.5 aB |
| | Twice | 53.5 bA | 57.1 bA | 69.0 bA | 72.0 bA | 39.6 bB | 44.7 bA | 58.3 bB | 60.5 bA |
| | Thrice | 44.7 cB | 52.2 bA | 59.2 cA | 62.9 cA | 33.1 cA | 35.8 cA | 44.9 cA | 51.4 cA |
| | *p* > α | <0.001 | <0.001 | <0.001 | <0.001 | <0.001 | <0.001 | <0.001 | <0.001 |
| 2017 | Once | 60.1 aA | 56.6 aA | 75.4 aA | 71.2 aA | 39.2 aA | 38.0 aA | 71.2 aA | 65.4 aB |
| | Twice | 46.5 bA | 45.4 bA | 68.0 bA | 66.6 aA | 34.4 bA | 35.6 aA | 60.5 bA | 61.6 bA |
| | Thrice | 41.9 bA | 40.4 cA | 59.4 cA | 59.1 bA | 36.1a bA | 37.7 aA | 45.7cB | 52.2 cA |
| | *p* > α | <0.001 | <0.001 | <0.001 | <0.001 | 0.027 | 0.289 | <0.001 | <0.001 |
| 2018 | Once | 86.0 aA | 80.9 aA | 87.5 aA | 84.2 aA | 46.6 aA | 39.9 bB | 76.4 aA | 73.2 aA |
| | Twice | 61.8 bA | 58.8 bA | 74.6 bA | 75.3a bA | 44.7a bA | 41.9a bA | 61.8 bA | 61.2 bA |
| | Thrice | 52.7 cA | 56.0 bA | 67.7 bA | 70.4 bA | 42.3 bA | 44.5 aA | 46.0 cA | 50.1 cA |
| | *p* > α | <0.001 | <0.001 | <0.001 | <0.001 | 0.022 | 0.018 | <0.001 | <0.001 |
| | | Late-fall Sward Heights | | | | | | | |
| 2014 | Once | 171.5 aA | 159.6 aA | 70.9 aB | 96.3 aA | 197.9 aA | 173.1aB | 194.4 aA | 174.1 aB |
| | Twice | 69.8 bA | 61.1 bA | 64.5 aA | 65.5 bA | 69.2 bA | 63.5 bA | 91.0 bA | 95.4 bA |
| | Thrice | 47.4 cA | 49.9 bA | 49.9 bB | 57.3 bA | 66.2 bA | 58.5 bA | 85.6 bA | 86.2 cA |
| | *p* > α | <0.001 | <0.001 | <0.001 | <0.001 | <0.001 | <0.001 | <0.001 | <0.001 |
| 2015 | Once | 188.9 aA | 165.8 aB | 111.6 aA | 103.9 aB | 187.2 aA | 194.2 aA | 206.1 aA | 185.0 aB |
| | Twice | 165.0 bA | 145.5 bB | 82.7 bA | 87.2 bA | 157.9 bA | 160.0 bA | 152.2 bB | 158.9 bA |
| | Thrice | 30.1 cB | 40.1 cA | 45.3 cB | 50.9 cA | 96.5 cA | 90.2 cA | 57.5 cB | 66.7 cA |
| | *p* > α | <0.001 | <0.001 | <0.001 | <0.001 | <0.001 | <0.001 | <0.001 | <0.001 |
| 2016 | Once | 183.7 aA | 182.6 aA | 104.0 aB | 118.1 aA | 183.4aB | 194.6 aA | 208.5 aA | 205.7 aA |
| | Twice | 120.7 bB | 141.3 bA | 86.2 bB | 108.0 bA | 156.5 bA | 164.2 bA | 150.8 bA | 155.6 bA |
| | Thrice | 29.4 cB | 41.3 cA | 50.5cB | 69.9 cA | 92.6 cA | 110.1 cA | 54.7cB | 76.3 cA |
| | *p* > α | <0.001 | <0.001 | <0.001 | <0.001 | <0.001 | <0.001 | <0.001 | <0.001 |
| 2017 | Once | 186.7 aA | 167.9 aB | 90.8 aA | 91.0 aA | 179.6 aA | 168.7 aB | 207.9 aA | 207.6 aA |
| | Twice | 89.8 bA | 70.4 bB | 69.4 bA | 71.6 bA | 148.1 bA | 134.0 bB | 122.7 bA | 121.5 bA |
| | Thrice | 32.9 cB | 37.7 cA | 41.5 cA | 44.6 cA | 42.3 cA | 41.4 cA | 45.6 cA | 47.7 cA |
| | *p* > α | <0.001 | <0.001 | <0.001 | <0.001 | <0.001 | <0.001 | <0.001 | <0.001 |

NWSG = big bluestem—*Andropogon gerardii*, Gamagrass—*Tripsacum dactyloides*, Indiangrass—*Soghastrum nutans*, and Switchgrass—*Panicum virgatum*; † the height at which a horizontal meter stick held across a harvest strip touched at list three topmost leaves of the NWSG bunches; § Means of the same grass followed by the same lowercase letter within-, or uppercase letter between-columns are not statistically different at α = 0.05; # Probability of mean difference between harvest frequencies, within a year. Note: There was no treatment effect on sward-height records for May 2014, the same year the harvest frequencies were first imposed.

The sward-heights for these three-cut strips were shorter than their respective single-cut values by about 14% for IG, 26.5% for BB and GG and up to 43% for SG. The observed

shorter sward-heights for the three-cut strips was actually in agreement with reported aftermath reflection of severity and frequency of defoliation history in *Dactylis glomerata* stands [35]. Similar responses to defoliation are also reported from a grazing trial on elephant grass (*Pennisetum purpureum*) with regrowth sward-heights being shorter for the severely grazed stands compared to those grazed lightly [36]. As well, regrowth in the more severely grazed stands (70% defoliation) showed reduced leaf appearance, lower stem and leaf elongation rates, and higher proportions of sheaths than leaf lamina, unlike those defoliated at 50%. According to [37], leaf sheaths contribute less than 5% of canopy photosynthesis, which may explain the associated lower rates of leaf elongation in the more severely defoliated stands. In the respective two-cut strips, the swards were significantly taller ($p < 0.001$) than in the three- but shorter than the single-cuts. However, in the IG plots, differences between the two- and single-cut sward-heights were not significant ($p > 0.05$). This indicates that, for the two-cut forage harvest systems, the early-season growing conditions were good enough for the NWSG stands to compensate for differences in severity of the prior-year defoliation tissue-damages. The timing and frequency of prior-years defoliation often reflect in aftermath growth performance including DM yield, bud and tiller numbers for the NWSGs of North America [16,38]. Up to 60% greater herbage yields have been reported from unclipped wester wheatgrass (*Pascopyrum smithii*, Rydb.) compared to their counterparts subjected to multiple defoliations [39]. For the IG, however, the single- and two-cut strips had comparable sward-heights as the three-cuts in BB plots implying that its early-season resources were probably diverted to other growth components rather soon. That also suggests that its relatively more spread crowns were less prone to self-shading and, effectively, their tillers attained dependable photosynthetic capacities rather faster.

After the 2015 flipping of harvest frequencies between the single- and three-cut strips, the early-season sward heights recorded in summer of 2016, (Table 4), tended to be consistently greater in the three-cut strips that were previously cut once year$^{-1}$ than those cut thrice all along, although the differences were only significant for BB. In the same assessment, sward-heights in the corresponding single-cut strips that flipped from three cuts year$^{-1}$ were statistically similar to those cut only once each year, except in SG plots. Growth performance of the flipped single-cut strips, through to the 2018 assessment, also exhibited the ability to compensate for prior losses in growth vigor, except in IG plots (Figure 3). For IG, the corresponding early-summer sward height values only came close to, but never reached 50 cm. Such species differences in response to shared defoliation management are often attributable to respective morphological and physiological characteristics that can translate into different regrowth rates [34]. Similar differences in species compensatory dynamics, in a plant community, have been credited to increases in the abundance of one species with a decrease in another [27]. By the 2018 harvest-year and within species, the 2014 single-cut strips that in 2015 flipped into three cuts year$^{-1}$ were statistically the same height as those harvested thrice all along ($p < 0.001$). However, numerically, the flipped strips maintained an edge over the ones that were not. The observed growth responses to the change in harvest regimes indicate that flipping the harvest frequencies can strategically induce conducive growth responses for both forage biomass and sward structures of managed NWSG stands. In fact, in both the two- and three-cut strips, the mean sward-heights exceeded the 30–40 cm minimum grass height requirement for good ground-nesting bird habitat [28]. The single- and two-cut strips will also provide good winter cover for a variety of wildlife, including small mammals. A practical flipping-of-harvest-regimes scenario could involve dividing a NWSG field into halves where one section is mowed thrice year$^{-1}$, for hay, and the other only once year$^{-1}$ as feedstock in alternate years. During the summer, small mammals and grassland birds can forage and find cover in the three-cut stands while the single-cut year$^{-1}$ section may also provide forage for herbivores such as the white-tailed deer (*Odocoileus virginianus*).

### 3.2. Late-Fall Sward-Heights

Also, in Table 4, are the results of a season-end assessment of the effect of changing harvest frequencies on the NWSG aftermath sward structures. Except in IG plots, the sward-heights were significantly taller ($p < 0.05$) in all three-cut strips that were previously cut once year$^{-1}$ than those receiving three cuts year$^{-1}$ since 2014. In 2015, the late-fall sward-heights were about 6-, 9-, and 10-cm taller for the flipped three-cut strips in GG, SG, and BB plots, respectively, than their non-flipped counterparts. The observed superiority in late-fall sward-heights of the flipped three-cut strips was actually clearer in the 2016 when values were from 27% greater for BB to 29% for both GG and SG, respectively. For the same species, the 2017 late-fall swards still tended to be taller in the flipped three-cut strips than in those cut thrice for three consecutive years, but only significantly so in BB plots ($p < 0.001$). Usually, multiple cuts per year remove apical dominance and may encourage tillering [40–42] and also reduce root growth and root reserves [42]. Repeated cutting is reported to reduce tiller structural carbohydrate contents, which has an effect on tiller growth vigor [43]. The greater height in the plants flipped from single-cut may be attributed to better root reserves that allowed more vibrant growth compare to those in a continuous three-cut year$^{-1}$ regime. Also, there is a potential that the plants transitioning to a single-cut from a three-cut year$^{-1}$ had fewer tillers hence reduced competition for resources that allowed them better overall growth.

Among the strips cut only once year$^{-1}$, the 2015 late-fall sward-heights were 8–23 cm taller in the strips that flipped from the three-cut year$^{-1}$ regime. In fact, the relative increases in late-fall sward-heights between 2015 and 2016 also showed abilities for the NWSG stands to compensate for tissue-damages sustained during the three-cut year$^{-1}$ harvest regime. For the strips that were maintained on the same harvest regime, the increases in mean late-fall sward-heights were only about 12 and 17 cm for SG and BB, respectively, and not more than 41 cm for GG. In comparison, the sward-heights for strips that flipped from three- to a single-cut year$^{-1}$ regime equaled 2–3-fold increases from ~50 cm to ~86 cm in 2014 to the range 104–206 cm in 2015. However, a within-year comparison between the growth performances of the flipped and not flipped single-cut strips showed that, except for IG, a single year's rest is not enough for the flipping effect to close the respective sward-height differences. As evidenced here, following two rest-years, the 2016 late-fall sward-heights of all strips whose harvest frequencies were flipped from three- to single-cut year$^{-1}$ in 2015 matched or exceeded their counterparts, significantly ($p < 0.05$). During the first year upon transitioning to one- from a three-cut per year, the plants had enough time to build carbohydrate reserves in their tiller buds and roots. Such reserves serve to promote prolific regrowth and sustained growth later as previously reported [43].

### 3.3. Basal Diameter

The assessment of growth responses of the NWSG stands to defoliation management was also based on crown/basal diameters (BD) recorded in early-spring whose results are summarized in Table 5. There was significant year × species × harvest regime interactions affecting the measured BDs and so the results are discussed separately, for each factor. As in Figure 4, the mean BD for each species were generally greater in 2016 than in 2015. Those harvested thrice year$^{-1}$ had BDs with lower magnitudes and which were significant ($p = 0.05$), in most cases. This agrees with previous findings where more intensely grazed pastures showed reduced basal areas [44]. An earlier study on grass growth under grazing found that periodic heavy grazing during the growing season restricted basal-area growth [45]. In a bunch-grass clipping frequency trial with *Arrhenatherum elatius*, [21], data showed that the crown diameters of the less frequently defoliated plants were greater than those defoliated more frequently. Similarly, it is reported that grazing up to 80% forage biomass removal resulted in significant reduction in live basal cover [46]. Within species, the BD values tended to be greater in the strips whose harvest frequencies were flipped than those harvested the same, consecutively. However, the observed flipping effects were only significant among the three-cuts in SG plots (Table 5). Within a harvest regime, the

BD differences due to harvest frequencies were not in any consistent pattern and mostly numerical, except for the flipped strips in BB plots. The observed differences in species response to the harvest regimes agree with earlier reports [34] that attributed responses to species morphological and physiological characteristics.

**Table 5.** Effects of a seasonal change (Same vs. Flipped) in harvest regimes † on early-spring and -summer basal and canopy diameters and their ratios §, respectively, in native warm-season grass stands harvested once, twice or thrice in the first year with or without a frequency switch between the one- and three-cut strips during the second-year.

| Year | Cuts | Species and Harvest Regime | | | | | | | |
|---|---|---|---|---|---|---|---|---|---|
| | | Big Bluestem | | Gamagrass | | Indiangrass | | Switchgrass | |
| | | Same | Flipped | Same | Flipped | Same | Flipped | Same | Flipped |
| | | Basal | | | | | | | |
| | | -------------------------------------------cm------------------------------------- | | | | | | | |
| 2015 | Once | 21.7 a | 26.2 a | 19.1 aA | 19.8 bA | 19.4 aA | 21.3 aA | 18.6 aA | 21.0 aA |
| | Twice | 17.4 aA | 18.5 bA | 19.4 aB | 29.8 aA | 18.4 aA | 22.7 aA | 17.3 aA | 18.8 aA |
| | Thrice | 17.2 aA | 19.6 bA | 24.4 aA | 28.7 aA | 19.4 aA | 21.8 aA | 17.2 aB | 21.1 aA |
| | *p* > α | 0.134 | <0.001 | 0.257 | <0.001 | 0.853 | 0.828 | 0.541 | 0.455 |
| 2016 | Once | 26.8 aA | 27.6 bA | 39.7 aA | 36.9 aA | 28.1 aA | 30.2 aA | 25.1 aA | 28.1 aA |
| | Twice | 28.2 aA | 29.5 abA | 37.4 aA | 38.6 aA | 30.2 aA | 32.2 aA | 27.5 aA | 29.4 aA |
| | Thrice | 28.3 aA | 31.3 aA | 37.9 aA | 37.5 aA | 32.2 aA | 34.0 aA | 26.4 aB | 31.1 aA |
| | *p* > α | 0.718 | 0.052 | 0.783 | 0.727 | 0.231 | 0.157 | 0.244 | 0.169 |
| | | Canopy | | | | | | | |
| | | -------------------------------------------cm------------------------------------- | | | | | | | |
| 2015 | Once | 46.3 aA | 39.8 aA | 27.7 bA | 30.3 aA | 20.1 bA | 21.8 bA | 29.1 aA | 24.6 aB |
| | Twice | 37.0 abA | 32.5 bA | 33.3 aA | 34.2 aA | 26.5 aA | 25.1 bA | 18.7 bA | 19.5 bA |
| | Thrice | 35.3 bA | 30.6 bA | 37.0 aA | 31.7 aA | 28.8 aA | 30.1 aA | 21.1 bA | 24.2 aA |
| | *p* > α | 0.101 | 0.035 | 0.257 | 0.332 | 0.022 | 0.007 | <0.001 | 0.007 |
| 2016 | Once | 57.1 aB | 63.3 aA | 63.7 abB | 81.6 aA | 52.8 bA | 55.6 aA | 55.2 aA | 58.5 aA |
| | Twice | 58.9 aA | 60.2 aA | 70.1 aA | 78.1 aA | 60.5 aA | 58.0 aA | 52.0 aB | 59.3 aA |
| | Thrice | 49.0 bA | 53.1 bA | 56.2 bB | 70.0 bA | 52.7 b | 52.5 a | 44.9 bB | 50.1 bA |
| | *p* > α | <0.001 | 0.002 | 0.009 | <0.001 | 0.069 | 0.310 | 0.001 | <0.001 |
| | | Canopy: Basal Diameter | | | | | | | |
| | | -------------------------------------------Ration------------------------------------- | | | | | | | |
| 2015 | Once | 2.40 aA | 1.56 aB | 1.49 aA | 1.59 aA | 1.09 bA | 1.04 aA | 1.61 aA | 1.23 aB |
| | Twice | 2.23 aA | 1.78 aA | 2.08 aA | 1.17 bA | 1.48 aA | 1.19 aA | 1.10 bA | 1.09 aA |
| | Thrice | 2.14 aA | 1.57 aB | 1.59 aA | 1.11 bB | 1.50 aA | 1.45 aA | 1.24 bA | 1.16 aA |
| | *p* > α | 0.858 | 0.408 | 0.316 | 0.005 | 0.038 | 0.141 | 0.008 | 0.655 |
| 2016 | Once | 2.14 aA | 2.31 aA | 1.63 abB | 2.23 aA | 1.93 aA | 1.85 aA | 2.22 aA | 2.10 aA |
| | Twice | 2.10 aA | 2.05 aA | 1.90 aA | 2.05 abA | 2.05 aA | 1.84 aA | 1.91 bA | 2.04 aA |
| | Thrice | 1.80 bA | 1.72 bA | 1.52 bB | 1.88 bA | 1.67 aA | 1.55 aA | 1.71 bA | 1.62 bA |
| | *p* > α | 0.054 | 0.001 | 0.045 | 0.021 | 0.163 | 0.112 | 0.007 | <0.001 |

NWSG = big bluestem—*Andropogon gerardii*, Gamagrass—*Tripsacum dactyloides*, Indiangrass—*Soghastrum nutans*, and Switchgrass—*Panicum virgatum*; † harvesting once, twice or thrice a year with or without a second-year switch between the single and three cuts assignments. § horizontal distance between opposite outer edges of the clump base or its canopy of the same NWSG bunch, average of three each. The same lowercase letter within-, or uppercase letter between-columns are not statistically different at α = 0.05

### 3.4. Canopy Diameter

To assess how defoliation management might be reflected in vegetation structural responses, changes in species CD values is also helpful. In the current study, the early-summer CD values showed significant year × species × harvest regime interactions and so the results are presented separately by year, species and harvest regimes (Table 5). In most

cases, the NWSG clumps cut once year$^{-1}$ had greater CD values than their counterparts cut twice or thrice year$^{-1}$. A similar observation with 20% greater CD in ungrazed bunchgrasses compared with their grazed counterparts is reported [47]. In the current study, each species also registered significantly greater CD values in 2016 than in 2015 (Figure 4). This observation might be linked to progressive growth performance among the grasses as the stand matures. From the 2015 data, differences due to prior-year harvest frequencies were only significant in SG plots, among the single-cut strips, with those previously cut once year$^{-1}$ performing better than their three-cut counterparts. In 2016, however, plants in the strips that flipped from three- to a single-cut year$^{-1}$, exhibited significantly greater CDs in BB and GG, but not IG or SG plots. Among the strips cut thrice year$^{-1}$, the same was true in GG and SG, but not BB or IG plots. Similar differences in species characteristics and physiological and morphological responses to frequency and intensity of defoliation are reported [34].

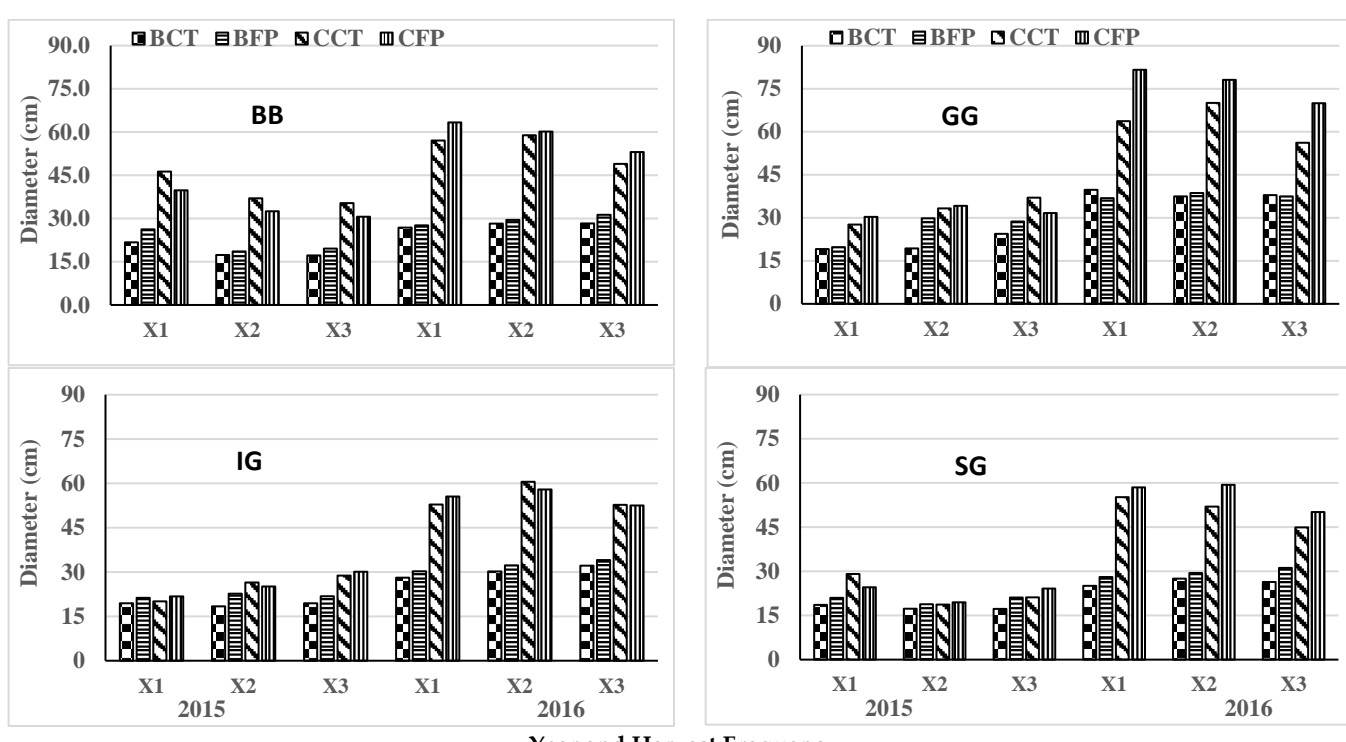

**Figure 4.** Early-spring basal- (B) and canopy (C) diameters of native warm-season grass [(BB) big bluestem—*Andropogon gerardii*, (GG) Gamagrass—*Tripsacum dactyloides,* (IG) Indiangrass—*Soghastrum nutans,* and (SG) Switchgrass—*Panicum virgatum*] clumps as affected by being harvested once—X1, twice—X2, or thrice—X3 year$^{-1}$, continuously (CT) or with the harvest regimes for the X1 and X3 harvest strips flipped during the second year as recorded in 2015 and 2016. Note: The 32 plots that did not have harvest regimes flipped were in a separate block from the other 32 that did.

Although the recorded 2016 mean CD values were consistently greater than in 2015, the data also showed species differences due to the flipping of harvest frequencies (Figure 4). Unlike the BD values, the 2015 CDs differed significantly between the single- and three-cut strips, except in the GG plots where cut frequencies flipped. Also, it is worth noting the species difference in the recorded CD, and that, in 2015, IG bunches in the single-cut strips had smaller CDs than those cut thrice year$^{-1}$. Significant CD differences between the single- and three-cut strips were also observed in 2016 except in the IG plots in which harvest frequencies were flipped. Stems under the one-cut year$^{-1}$ regime were taller and more outward leaning plus, though not determined, had leaves that appeared longer and broader. These may have contributed to the observed differences in CD between harvest regimes. That is so because swards submitted to more severe grazing often have higher proportions

of sheaths than leaves lamina while, in pastures with 50% defoliation, higher leaf-area and leaf elongation rates are more likely [37]. Others studies assessing defoliation effects on range species have reported 60% higher herbage yields by unclipped western wheat grass (*Pascopyrum smithii*, Rydb.) than their counterparts receiving multiple defoliations [39].

### 3.5. Canopy-to-Basal Diameter Ratios

While changes in BD, in young NWSG stands, indicate dynamics in tiller numbers, the regrowth CDs may provide information on how the former relates to species proportional ground cover between harvest events. The statistical analysis of the within-treatment CD:BD ratios showed species and year differences ($p < 0.004$) in proportional CD and BD responses to the changes in harvest regimes (Table 5). In 2015, only the BB clumps in the strips harvested the same, consistently, had canopies that were twice as wide as their crowns. However, in 2016, that was true for all single-cut BB and GG strips and their corresponding SG strips with prior three-cut year$^{-1}$ experience. Also, in 2015, and within species, canopies in the single-cut BB and SG clumps outsized their crowns in the strips harvested the same all along than those flipped from three-cuts. Still, in 2016, differences among the single-cut strips were only significant in GG plots where clumps in the strips that were previously cut thrice year$^{-1}$ had greater CD:BD ratios than those cut once only. Besides, year differences in growing conditions and timing of operations, the observed similarity in CD:BD ratios may be attributable to compensatory NWSG growth responses in the strips with prior multiple defoliation experience. Similar increases in CDs for less frequently defoliated bunch grasses are reported [21].

On the current data (Table 5), the higher CD:BD ratios also indicate preferential resource allocation for the expansion and elongation of residual leaf and stem tissues relative to the growth of new tillers. The dissimilarities in CD:BD ratios are attributable to species differences in compensatory growth responses to the change in defoliation regimes. That often involves changes in respiratory rates, growth rates, and carbon allocation patterns [3]. Specific responses to defoliation regimes are rooted in morphological adaptations to physical tissue-damages, which increase plants' tolerance to defoliation [12,13,15]. By large, these species growth responses to defoliation, strategically seek to repair the tissue-damages, and restore lost physiological functions. That often involves increased tiller density in the more frequently harvested stands and reduced vegetative growth on the less frequently harvested ones [17,38,48]. Depending on the severity of sustained injuries, recovering NWSGs usually exhibit preferential allocation of energy reserves to replace lost leaf area and/or increase photosynthesis rates on the residual or regrowth tissues [17,48].

Year and species differences in CB:CD ratios were also observed among the three-cut strips. In 2015, the CB:CD ratios tended to be greater in the strips harvested thrice every year than those previously harvested only once year$^{-1}$. However, significant differences were only present in the BB and GG plots. In 2016, however, the BD:CD difference was only significant in GG plots where the flipped previous single-cut strips outnumbered those harvested thrice year$^{-1}$, consistently. Species morphological differences may explain the demonstrated ability for GG, whose stems are more prostrate oriented and with larger proportion of growing points more likely to escape defoliation physical tissue-damages, to be less impacted by frequent harvesting. This agrees with previous reports on insensitivity of GG to frequent defoliation [49]. The portrayed species differences in compensatory growth potentials underscore the importance of species-specific defoliation management.

### 3.6. Sward Structure

The effects of flipping the harvest frequencies on subsequent NWSG performance were also assessed based on changes in sward structural features associated with canopy light interception and stand density. The proportions of instantaneous PAR above the canopy intercepted by the vegetation layer, in 2015 and 2016, are summarized by harvest regimes, within species (Table 6). In the same table, mean VOH, within-species and for the same harvest regime are presented.

**Table 6.** Effects of a seasonal change (Same vs. Flipped) in harvest frequency (Cuts) on subsequent early-season canopy closure recorded in May 2015 and 2016, and late-season stand density of young NWSG stands based on respective light interceptions † and visual obstruction heights § recorded in October 2016 and 2017.

| Year | Cuts | Species and Harvest Regime | | | | | | | |
|---|---|---|---|---|---|---|---|---|---|
| | | **Big Bluestem** | | **Gamagrass** | | **Indiangrass** | | **Switchgrass** | |
| | | Same | Flipped | Same | Flipped | Same | Flipped | Same | Flipped |
| | | Light Interception | | | | | | | |
| | | ------------------------------------------%------------------------------------------ | | | | | | | |
| 2015 | Once | 93.5 aA * | 93.0 aA | 79.6 aB | 89.0 aA | 76.7 aA | 82.0 aA | 79.2 aA | 84.4 aA |
| | Twice | 86.2 bA | 83.6 bA | 66.9 bB | 80.8 bA | 67.0 bA | 73.0 bA | 48.1 bB | 62.1 bA |
| | Thrice | 81.2 bA | 79.4 bA | 58.2 cB | 73.9 cA | 55.5 bA | 62.0 cA | 38.9 cB | 55.4 bA |
| | *p* > α # | <0.001 | 0.011 | 0.007 | <0.001 | 0.003 | <0.001 | <0.001 | <0.001 |
| 2016 | Once | 94.8 aA | 93.1 aA | 89.8 aA | 88.7 aA | 85.9 aA | 89.9 aA | 93.3 aA | 92.5 aA |
| | Twice | 85.5 bA | 81.6 bA | 75.3 bA | 77.0 bA | 80.3 bA | 78.4 bA | 74.3 bA | 71.2 bA |
| | Thrice | 77.2 cA | 81.6 bA | 65.4 cB | 75.9 bA | 74.3 cA | 78.3 bA | 62.2 cA | 68.0 bA |
| | *p* > α | <0.001 | 0.009 | <0.001 | <0.001 | <0.001 | 0.002 | <0.001 | <0.001 |
| | | Visual Obstruction Height | | | | | | | |
| | | ------------------------------------------cm------------------------------------------ | | | | | | | |
| 2016 | Once | 186.3 aA | 169.7 aA | 81.4 aA | 78.7 aA | 181.9 aA | 172.9 aA | 208.7 aA | 210.0 aA |
| | Twice | 73.5 bA | 85.0 bA | 69.4 bA | 71.2 aA | 111.4 bA | 132.8 bA | 136.5 bA | 148.9 bA |
| | Thrice | 21.3 cA | 18.0 cB | 35.3 cA | 35.0 bA | 29.6 cA | 28.9 cA | 32.7 cB | 41.4 cA |
| | *p* > α | <0.001 | <0.001 | <0.001 | <0.001 | <0.001 | <0.001 | <0.001 | <0.001 |
| 2017 | Once | 176.0 aA | 172.3 aA | 93.1 aA | 72.0 aA | 166.1 aA | 173.7 aA | 205.4 aA | 210.0 aA |
| | Twice | 73.7 bA | 56.6 bB | 67.7 aA | 58.5 bA | 120.1 bA | 110.4 bA | 102.1 bA | 110.1 bA |
| | Thrice | 29.9 cA | 26.9 cA | 34.2 bA | 35.3 bA | 35.1 cA | 28.1 cB | 34.5 cA | 39.4 cA |
| | *p* > α | <0.001 | <0.001 | 0.002 | <0.001 | <0.001 | <0.001 | <0.001 | <0.001 |

NWSG = big bluestem—*Andropogon gerardii*, Gamagrass—*Tripsacum dactyloides,* Indiangrass—*Soghastrum nutans,* and Switchgrass—*Panicum virgatum*; † The proportions of photosynthetically active radiation measured above the canopy that did not reach the ground vertically below the canopy; § The height at which a white card held against a Robel pole was invisible, through the vegetation, to the naked eye positioned about a meter above ground; * Means of the same grass followed by the same lowercase letter within-, or uppercase letter between-columns are not statistically different at α = 0.05. # Probability of mean difference between harvest frequencies within a year.

### 3.6.1. Canopy Light Interception

As expected, light interception values recorded in 2015 were greater for the single cuts than their respective two- and three-cuts year$^{-1}$ (Table 6). The fact that the percentage PAR interception in all flipped single-cut strips either matched or slightly exceeded that of their non-flipped counterparts is an indication of the positive impact of allowing plants long recovery rests to repair their damaged tissues. Species differences in the ability to restore the lost photosynthetic surface may explain the significantly greater values, nearly 10-points margin, observed among the single-cuts in the GG plots. Likewise, for GG and SG, the percentage PAR interception, among the three-cut strips, were statistically greater for the strips that had previously received a single-cut year$^{-1}$ than those on a second three-cut year $^{-1}$ cycle. Logically, the previous three cuts year$^{-1}$ may have stimulated more tiller buds and, that lead to greater stem densities. That seems the most plausible scenario because, under such an open stand, the red/far-red ratio also reported to increase tiller/stem formation remains high [50]. And as the system transitions from a three- to a single-cut, the increase in tiller numbers due to previous defoliation management allowed for a more robust plant growth that resulted in an improved canopy structure and increased PAR interception. The observed changes can be attributed to the defoliation effects on canopy density that can profoundly influence the quality of radiation and the irradiant flux density received by the plants [51]. As roots recovered from effects of repeated defoliation in

previous years, there was potentially greater root density and better uptake of soil water and nutrients, which allowed for fast growth. In similar patterns, the 2016 early-season values for PAR interception mostly outnumbered their respective 2015 records, but differences due to previous defoliation experiences were only significant among the three-cut GG strips. The observed similarities in canopy closure attributes among the single- or three-cut strips that experienced the same or flipped harvest regimes, as indicated by larger PAR values, may also have resulted from a gap-filling effect of annual weeds. The early-season stands often have high proportions of annual weeds, which also include tall-growing broadleaf ones occupying voids between the bunch grasses, as reported in [30]. With respect to wildlife habitat quality features, bunched grasses interspersed with legumes and forbs with insect-attracting wild flowers are among desirable attributes of a good habitat for bobwhites [29].

### 3.6.2. Stand Density

Mean sward structural response assessment of the NWSG stands to the seasonal changes in harvest frequencies was also based on season-end VOH measurements in 2016 and 2017. There was significant year × species × frequency interaction on the measured VOH (Table 3). The results are presented and discussed separately (Table 6) by year, species and harvest regimes. Among the single-cut strips and, both in 2016 and 2017, VOH within species showed no effect due to the flipping of harvest frequencies ($p > 0.05$). The 2016 values averaged 178, 80, 177, and 209 cm for BB, GG, IG and SG, respectively, with their corresponding 2017 averages as ~174, 82, 170, and 208 cm. As an indicator of stand density, the recorded VOH values also seemed more reflective of the NWSG stem densities and leafiness than the presence of annual weeds. That may also explain their notable closeness, in trend, to the respective late-fall sward-heights (Table 4). The data also suggests that, in both years, the single-cut strips that were previously cut thrice, somehow compensated for their losses in stand vigor and hence the lack of difference due to prior harvest regimes. This better performance of the flipped three-cut strips against their non-flipped counterparts are consistent with the asserted prior-year defoliation-induced increases in tiller buds coupled with greater buildup of energy reserves [30], as the frequency changes from three- to a single-cut year$^{-1}$. In plant communities, compensatory dynamics are often manifested in different ways and, may involve increases in the abundance of one species at the expense of another [27]. In more frequently harvested stands, compensatory growth responses often result in increases in tiller density and delayed preferential allocation of resources to root growth [17,48]. That will logically boost the regrowth stand density, as reflected in the recorded VOHs, in the current study.

In assessing the NWSG structural response to the changes in harvest regimes, the three cut VOH values were also compared (Table 6). During the 2016, clear three-cut VOH differences due to prior harvest frequencies were detected in the BB (21.3 vs. 18.0 cm) and SG (32.7 vs. 41.4 cm), but not the GG (35.1 cm) or IG (29.2 cm) data. While the BB strips flipped from a single-cut year$^{-1}$ trailed their not-flipped counterparts in VOH, the opposite was true for SG. During the 2017, a flipping effect, among the three-cut strips was only significant in the IG (35.1 vs. 28.1 cm) plots. For the 2017 data, the three-cut VOH values were 3- and 5-cm greater in the not-flipped BB and the flipped SG strips, respectively. However, while these differences observed for BB and SG were insignificant, the 7-cm difference for IG was significant. Largely, the dissimilarities in the recovery stand density originate from differences among species in the susceptibility of shoot growing points to physical damages and how they, preferentially, allocate resources to new tillers vs regrowth on residual tissues. Likewise, yearly differences in growing conditions can impact the proportions of annual weeds in the stands. These differential structural responses to defoliation management have implications on such ecosystem services as ground cover, wildlife habitat quality—food and shelter, and airflow through the stands. Depending on the species composition and the desired season-end ecosystem services, appropriate strategies on the frequency and timing of harvesting operations can be implemented.

In both years, the season-end VOH means were about 2–3-folds greater ($p < 0.001$) in the single- than the three-cut GG strips, 9-folds so for BB in 2016 and five to six times in all other plots (Table 6). The VOH values for the single-cuts also outnumbered the two-cuts, significantly ($p < 0.001$), except in 2016 flipped and the 2017 not-flipped GG strips. All strips cut twice year$^{-1}$ still had significantly greater VOH values compared to their respective three-cut counterparts. These results are consistent with typical NWSG growth responses to defoliation associated with greater tiller densities in the absence of apical dominance [40,41]. Frequently harvested grasses often produce tillers with reduced structural carbohydrate contents, which negatively impacts their growth vigor [43]. Although the two-cut strips experienced no flipping of harvest frequencies, the 2016 single- and three-cut BB, IG, and SG stands in the flipped-strips block had, numerically, greater VOH values, about 10-units taller, than those in plots harvested the same, continuously. This better performance of the two-cut strips in the flipped-plots block is mostly attributable to the resulting better sunlight environment as they all bordered the relatively shorter three-cut strips to the North. In the not-flipped block, however, the matching two-cut strips would have experienced relatively more shading from the taller single-cut strips to their North. Considering also the temporal microclimate implications that this may have on the associated wildlife habitat attributes, the importance of field layout, orientation, and the timing of operations to strategic defoliation management of NWSG stands could not be clearer.

## 4. Conclusions

The data has shown that strategic defoliation management of NWSG stands for forage biomass and or feedstock production may have species-specific practical implications on subsequent wildlife habitat quality and other ecosystem services. The demonstrated differences in stand structural responses to the changes in harvest regimes also shows the importance of taking into consideration species inherent morphological and physiological adaptations to grazing, when planning sustainable defoliation management of mixed NWSG stands. The data also shows that each of the NWSGs has the potential for developing a multipurpose summer forage system that may also provide wildlife shelter from extreme weather conditions.

Based on the observed differences in the sward structural responses to defoliation (sward-heights, canopy closure, percent light interception and visual obstruction) due to species and changes in harvest regimes, strategic defoliation management of NWSG fields for both forage biomass production and wildlife habitat must aim at creating spatial-structural heterogeneity, which may involve partitioning the field into adjacent sections so one can be repeatedly mowed for forage while the other provides critical wildlife habitat needs.

The observed lack of effect due to flipping of the harvest frequencies on the regrowth sward-heights demonstrated the inherent ability for the NWSGs to exhibit compensatory structural responses to changes in defoliation management. The flipping effects on stand density and canopy closure as indicated by the mean PARi and VOH readings further underscored the importance of using a combination of response variables for evaluating potential impact of defoliation management practices on wildlife habitat quality and other ecosystem services.

**Author Contributions:** Conceptualization, visualization, supervision, project administration, funding acquisition, formal analysis, investigation, and writing—original draft preparation, V.W.T.; methodology, validation, writing—review and editing, V.W.T. and M.K.K. All authors have read and agreed to the published version of the manuscript.

**Funding:** This research was funded by USDA-NIFA through the EVANS ALLEN program.

**Data Availability Statement:** Not Applicable.

**Acknowledgments:** Throughout the study and during the manuscript preparation, the project team received tremendous administrative and logistical support from the management of the Agricultural Research Station (ARS) at Virginia State University for which the authors are very grateful. As well, the technical support of the plant science and small ruminant research technicians (Amanda Miller, Joshquinn Andrews, and Ryan Mason) for research plots maintenance and data collection and that of ARS research students (Bianca Jacques, Courtney Epps, Ariel Coleman, David Johnson, Christos Galanopoulos and Christopher Copeland), who took measurements and typed data, are acknowledged. This article is a publication No. 390 of the ARS at Virginia State University.

**Conflicts of Interest:** The authors declare no conflict of interest. The funders had no role in the design of the study; in the collection, analyses, or interpretation of data; in the writing of the manuscript; or in the decision to publish the result.

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
