# Peer review of "Compensatory Structural Growth Responses of Early-Succession Native Warm-Season Grass Stands to Defoliation Management"

_agronomy, doi:10.3390/agronomy13051280_

Round 1

Reviewer 1 Report

The sward responses to defoliation frequency treatments in four native warm season grass species provides useful insight into the management of those species for multiple goals. The switching of cutting treatments from frequently to infrequently and vis-versa is particularly interesting. 

However, unfortunately, the experimental design as described does not allow for statistical comparison of the ‘flipped’ and ‘same’ treatments, as those treatments were assigned systematically by row number rather than randomly across all experimental units. Comparison of those treatments, as presented in Table 4, is inappropriate given the assumptions of analysis of variance.

Additionally, the statistical analysis does not appropriately address the repeated measurements collected from plots across years, and as a result the degrees of freedom are inflated. The analysis also does not properly address the strip plot/split block design used to apply the cutting treatments (Fig 2 and lines 152-155). Separate error terms are required for the cutting systems treatment by block and the grass species by block. 

The authors should describe more clearly why certain vegetation measures were collected in certain years but not in others.  Explanation of why certain measures were collected at specific timepoints and perhaps a visual timeline of data collection would be useful.

Use of biased terminology, particularity “damage” or “injury” as related to defoliation should be omitted throughout the manuscript. 

In the methods section, further detail is required on the source and germplasm/varieties of the four species planted as well as the physical arrangement of the plants within the transplanted plots.

The measure “stand thickness” is not adequately described. Is this visual obstruction height? Stand thickness does not seem to be the appropriate term for this measure. Perhaps density rather than thickness

Line 23/24 – a word(s) may be missing from this sentence. Which plots had their cutting treatments flipped? This sentence makes it seem like the 2 and 3 cut strips were flipped while lines 151/152 indicate that the 1 and 3 cut strips were flipped.

Author Response

The sward responses to defoliation frequency treatments in four native warm season grass species provides useful insight into the management of those species for multiple goals. The switching of cutting treatments from frequently to infrequently and vis-versa is particularly interesting.

Your professionalism and the candid complements, are appreciated.

However, unfortunately, the experimental design as described does not allow for statistical comparison of the ‘flipped’ and ‘same’ treatments, as those treatments were assigned systematically by row number rather than randomly across all experimental units. Comparison of those treatments, as presented in Table 4, is inappropriate given the assumptions of analysis of variance.

 Reasons added to the M&M, L158-162.

Additionally, the statistical analysis does not appropriately address the repeated measurements collected from plots across years, and as a result the degrees of freedom are inflated. The analysis also does not properly address the strip plot/split block design used to apply the cutting treatments (Fig 2 and lines 152-155).

Separate error terms are required for the cutting systems treatment by block and the grass species by block. 

Misleading statements in the statistical analysis section were removed and more text added for clarity.

The authors should describe more clearly why certain vegetation measures were collected in certain years but not in others.

Reasons for taking measurements on the other parameters with two years after flipping has been added to the M&M – L188-191

 Explanation of why certain measures were collected at specific timepoints and perhaps a visual timeline of data collection would be useful.

 An explanation has been added to L195-197

Use of biased terminology, particularity “damage” or “injury” as related to defoliation should be omitted throughout the manuscript. 

As appropriate, the term has been specified and “physical tissue-damages

In the methods section, further detail is required on the source and germplasm/varieties of the four species planted as well as the physical arrangement of the plants within the transplanted plots.

 Details added to L140-143.

The measure “stand thickness” is not adequately described. Is this visual obstruction height? Stand thickness does not seem to be the appropriate term for this measure. Perhaps density rather than thickness

Explanation expanded and “thickness” replaced with the less confusing term “density”, as suggested.

Line 23/24 – a word(s) may be missing from this sentence. Which plots had their cutting treatments flipped? This sentence makes it seem like the 2 and 3 cut strips were flipped while lines 151/152 indicate that the 1 and 3 cut strips were flipped.

 Good catch!. Corrected

Reviewer 2 Report

The objective of the study was to assess canopy structural responses of big bluestem (BB, Andropogon gerardii Vitman), eastern gamagrass (GG, Tripsacum dactyloides L.), indiangrass (IG, Sorghastrum nutans L.). Nash), and switchgrass (SG, Panicum virgatum L.) stands to seasonal changes in harvest frequencies. The information presented is interesting, however, it is necessary to provide additional information about how data were analyzed to quantify the "compensatory structural growth".

Please see the comments in the attached file.

Author Response

The objective of the study was to assess canopy structural responses of big bluestem (BB, Andropogon gerardii Vitman), eastern gamagrass (GG, Tripsacum dactyloides L.), indiangrass (IG, Sorghastrum nutans L.). Nash), and switchgrass (SG, Panicum virgatum L.) stands to seasonal changes in harvest frequencies. The information presented is interesting, however, it is necessary to provide additional information about how data were analyzed to quantify the "compensatory structural growth".

Your complements are appreciated and as appropriate, suggested revisions were addressed and clarification provided in respective comments.

Please see the comments in the attached file.

Reviewer 3 Report

Nothing is stated about soil fertility and fertilization.  How was this managed. In particular how often was nitrogen fertilizer applied during the year to the different treatments.

unsure of why treatments were switched on plots 33-64 (lines 150-151). why is carryover affect not occurring.

Line 606 has misspellings.

Author Response

Nothing is stated about soil fertility and fertilization.  How was this managed. In particular how often was nitrogen fertilizer applied during the year to the different treatments.

unsure of why treatments were switched on plots 33-64 (lines 150-151). why is carryover affect not occurring.

Line 606 has misspellings.

Thank you for time. There was no fertilizer applied, throughout the study. A statement added to M&M. Other sections were extensively revised for clarity.

Round 2

Reviewer 2 Report

Dear authors,

I just checked the responses presented. Some questions were addressed in the revised version of the manuscrit. My main concern is just about the use o non usual therminologies in grassland science. I believe that a clear understanding of what is each variable could improve the citability of the paper:

Please see Allen et al. 2011. An international terminology for grazing lands and grazing animals, Grass and Forage Science.

Please see some definitions provided by the authors bellow:

2.3.1 Sward (n.). A population or a community of herbaceous plants characterized by a relatively short habit of growth and relatively continuous ground cover, including both above- and below-ground parts.

2.3.2 Canopy (n.). The above-ground parts of a population or community of forage plants. It may include both herbaceous and woody vegetation.

2.3.2.1 Canopy architecture (n.). The spatial distri- bution and arrangement of the constituent parts of the canopy. 2.3.2.2 Canopy cover (n.). The proportion of the ground area covered by the canopy when viewed vertically.

2.3.2.3 Canopy density (n.). The bulk density of the canopy (mass unit volume)1).

2.3.2.4 Canopy height (n.). The surface height of an undisturbed canopy or the compressed height of a canopy, normally measured from ground level.

2.3.3 Botanical composition (n.). The relative propor- tions of the plant components (species and morpholog- ical units) in a canopy above a defined sampling height, preferably ground level.

Based on that:

How canopy diameter can be measured?

How basal diameter can be measured?

Why use cuts instead harvest frequency? 

Is it flipling a system?

I just recommend the authors to follow standard therminologies OR include a term definitions in the current paper.

Beside this, I think the information is interesting and can be accepted for publication.
